# The Quantity and Biochemical Composition of Sap Collected from Silver Birch (*Betula pendula* Roth) Trees Growing in Different Soils

**Justas Mingaila [1], Dovilė Čiuldienė [2,*], Pranas Viškelis [3], Edmundas Bartkevičius [1], Vladas Vilimas [1] and Kęstutis Armolaitis [2]**

[1] Vytautas Magnus University Agriculture Academy, Faculty of Forest Sciences and Ecology, Institute of Forest Biology and Silviculture, Studentų str.11, LT-53361 Akademija, Kaunas District, Lithuania; justasmingaila@gmail.com (J.M.); edmundas.bartkevicius@vdu.lt (E.B.); vladas.vilimas@vdu.lt (V.V.)

[2] Institute of Forestry, Lithuanian Research Centre for Agriculture and Forestry, Liepų str. 1, LT-53101 Girionys, Kaunas District, Lithuania; kestutis.armolaitis@lammc.lt

[3] Institute of Horticulture, Lithuanian Research Centre for Agriculture and Forestry, Kaunas str. 30, LT-54333 Babtai, Kaunas District, Lithuania; biochem@lsdi.lt

\* Correspondence: dovile.ciuldiene@lammc.lt; Tel.: +37-061-537-105

**Abstract:** Birch sap is colourless or slightly opalescent and is traditionally drunk in spring. Currently, birch sap is becoming more important in the market sector as well as to pharmacy companies due to its biochemical composition and use in a wide variety of products. To extract good quality sap using birch resources in a sustainable way, there is a need to investigate the influence of the dendrometric parameters of birch trees and soil properties on the quantity and chemical composition of birch sap. This study is performed in five silver birch (*Betula pendula* Roth) forest stands growing in Histosol, Luvisol and Arenosol with different moisture and nutrient contents. The results indicated that the most productive silver birch trees for sap harvesting were taller than 28 m, had a diameter at breast height over 40 cm and a crown base height greater than 19 m. Additionally, the highest quantity of birch sap was harvested from trees growing in well-aerated mineral soils (Arenosol and Luvisol) with normal moisture content. However, the sweetest birch sap was harvested from trees growing in nutrient-rich organic (undrained peatland Histosol) and temporarily flooded mineral (Luvisol) soils.

**Keywords:** *Betula pendula*; birch sap; biochemical composition; Histosol; Luvisol; Arenosol

## 1. Introduction

Birch sap is a non-timber forest product that has become more important from an economic and recreational point of view [1]. Birch sap is a traditional beverage in boreal and hemiboreal regions of the northern hemisphere [2–4]. Currently, the harvesting of birch sap remains an important activity mostly in Belarus, Estonia, Finland, Latvia, Lithuania, Poland, Romania, Russia and Ukraine due to the widespread distribution of birch species and the incorporation of sap into the former Soviet economic system [1]. However, birch sap is becoming more important in the market sector as well as to pharmacy companies of the European Union. Birch sap is used to manufacture health products, such as birch sap drinks flavoured with fruits, birch syrup and cosmetics for skin and hair. Furthermore, scientists from Poland have started to develop nonperishable, unpasteurized birch sap-based beverages, which can be classified as a superfood product [5].

Birch sap is a drink that is traditionally consumed in the spring. It is a colourless or slightly opalescent, scentless liquid with a slightly sweet mineral water taste due to the small amount of carbohydrates and the considerable amount of dissolved minerals it contains [6]. Viškelis and

Rubinskienė [7] reported that birch sap may contain soluble sugars, ascorbic acid, phenolic compounds, and micro- and macroelements, of which potassium was the predominant component.

In the boreal and hemiboreal regions, birch sap is directly tapped from native birch species, mainly from silver birch (*Betula pendula* Roth) and downy birch (*Betula pubescens* Ehrh.) [6,8–10]. According to the Global Forest Resource Assessment [11], these birch species account for 11%–16% of the total volume of forest stands in Russia and Fennoscandia, approximately 17% in Lithuania and 24%–28% in Belarus and Latvia. However, the volume of silver birch and downy birch accounts for only 1%–5% of the total volume of forest stands in Central Europe [11]. Silver birch grows in fertile forest sites with adequate moisture and air content and prefers sandy and silty tilled soils as well as fine, sandy soils [4,9]. Downy birch is predominant in wet, cool, fine-textured and poorly aerated soils.

The ability to harvest birch sap depends on geographical location and climate conditions. In boreal forests, the best time for birch sap harvesting is from the beginning of March to early April. However, birch sap harvesting starts later in northern countries (e.g., northern Russia and Finland) [3,8,12–14]. The increasing air temperature in the spring influences the metabolic level of living wood cells and thus affects the osmotic pressure of the water within the wood [15].

Numerous studies have confirmed that the quantity of harvested birch sap, as well as its physical and chemical properties, mostly depend on the birch species and the dendrometric parameters and can change during the harvesting period [8,16–19]. However, knowledge of the influence of soil on birch sap quantity and quality is scarce. Only two studies have investigated the chemical composition of birch sap harvested from stands growing in different soils [8,18]. Harvesting birch sap at industrial rates has started only in recent years in Lithuania. To extract only good quality sap using birch resources in a sustainable way, there is a need to investigate the influence of the nutrient status of different soils on the quantity and chemical composition of birch sap.

The aim of this study is to investigate the differences in the quantity and chemical properties of sap extracted from silver birch (*Betula pendula* Roth) forest stands growing in three different soils with different moisture and fertility conditions. The main objective of this study is to evaluate how the dendrometric parameters of studied pure silver birch stands and soil chemical properties influence sap quantity and biochemical composition.

## 2. Materials and Methods

### 2.1. Study Sites

The study was carried out in 2017 in forest stands of silver birch (*Betula pendula* Roth) growing in Histosol (Hs), Arenosol (Ar) and Luvisol (Lv) [20] (Figure 1, Table 1).

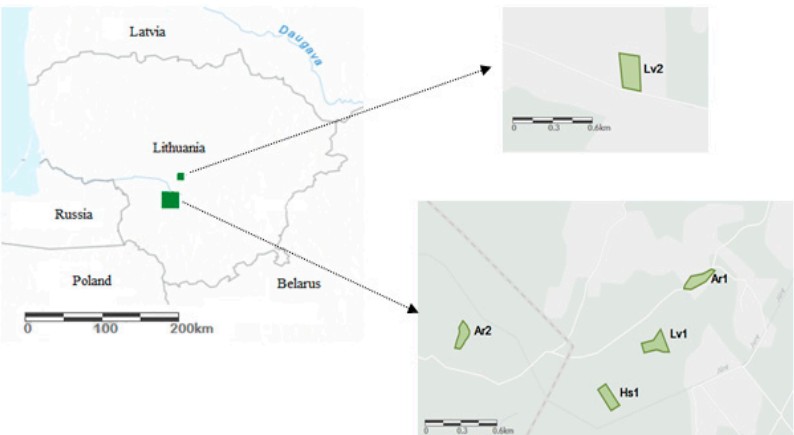

**Figure 1.** The studied silver birch (*Betula pendula* Roth) forest stands in the Kaunas region of the Lithuanian state forest enterprise. Coordinates: Ar1 (54°50′ N, 23°36′ E); Ar2 (54°50′ N, 23°38′ E); Lv1 (54°50′ N, 23°38′ E); Lv2 (55°2′ N, 23°50′ E); Hs1 (54°50′ N, 23°37′ E).

The studied pure silver birch stands growing in plains that are situated at 97 to 113 m a.s.l. and the area of the stands ranged from 0.5 to 2.3 ha. The climate in Lithuania is described as semi-humid and transitional between maritime and continental climates [21]. The average annual temperature is 7–7.5 °C, and the average annual precipitation is 650 mm. The main characteristics of the study sites are presented in Table 1.

**Table 1.** The mean dendrometric properties of pure silver birch (*Betula pendula* Roth) stands growing in different soils.

| Site | Age (yr) | Tree Density (ha$^{-1}$) | DBH (cm) | H (m) | Volume (m$^3$ ha$^{-1}$) | Soil Group [20] | Soil Moisture Condition |
|------|----------|--------------------------|----------|-------|--------------------------|-----------------|-------------------------|
| Hs1  | 73  | 175 | 38.8 ± 2.9 | 24.9 ± 1.3 | 225 | Histosol | Undrained peatland |
| Ar1  | 88  | 268 | 31.7 ± 2.3 | 28.5 ± 0.7 | 263 | Arenosol | Temporarily flooded mineral soils |
| Lv2  | 91  | 308 | 39.4 ± 1.2 | 25.0 ± 0.9 | 395 | Luvisol | |
| Ar2  | 105 | 223 | 35.6 ± 2.4 | 27.2 ± 0.7 | 265 | Arenosol | Mineral soils of normal moisture |
| Lv1  | 93  | 106 | 42.6 ± 2.7 | 28.5 ± 0.4 | 184 | Luvisol | |

**Note:** DBH—diameter at breast height; H—height of tree; CBH—crown base height. The results of DBH and H are presented as mean ± SE.

In this study, the silver birch forest stands were over-mature, and their age varied from 73 to 105 years. As seen from Table 1, the largest mean DBH (~43 cm) was found in the forest stand growing in Luvisol (Lv1) of normal humidity; however, the smallest mean DBH (~32 cm) was found in the forest stand growing in temporarily flooded Arenosol (Ar1). The largest mean tree height (~29 m) was found in stands growing in temporarily flooded Arenosol (Ar1) and Luvisol (Lv1) of normal humidity. Moreover, the smallest mean tree height (~25 m) was found in forest stands growing in undrained nutrient-rich peatland (Histosol, Hs1) and temporarily flooded Luvisol (Lv2). However, the dendrometric properties may also correspond to the different tree densities in the studied forest stands (Table 1).

### 2.2. Soil Sampling and Chemical Analysis

The soil sampling was carried out in October 2017. Composite soil samples (each at 3 systematically distributed points) were collected from the 0–20 and 20–40 cm deep layers (rhizosphere horizon) in three replicates along transects in each study site. The soil samples were taken with a 3 cm diameter metallic auger (in total, 90 samples were collected). From the soils samples, the following parameters were determined: the pH was analysed in a 1 M KCl suspension [22]; total nitrogen (TN) was found using the Kjeldahl method [23]; soil organic carbon (SOC) was found by dry combustion at 900 °C with a CNS analyser (Elementar Analy-sensysteme GmbH, Germany) [24]; mobile potassium (K$_2$O), mobile phosphorus (P$_2$O$_5$), mobile calcium (Ca) and mobile magnesium (Mg) were quantified by the égnér–Riehm–Domingo (A-L) method [25]. The pH parameters and chemical compositions of the studied soils are presented in Table 2.

The studied Histosol (site Hs1) had the highest ($p < 0.05$) concentrations of SOC and nutrients (total nitrogen (TN), mobile potassium (K$_2$O), Ca and Mg). The studied Arenosols (Sites Ar1 and Ar2) were rich in mobile phosphorus (P$_2$O$_5$) and, in contrast with the Histosol, had the lowest ($p < 0.05$) concentrations of SOC (only site Ar1), TN, Ca and Mg. The temporarily flooded Luvisol (Site Lv2) was characterised by having the highest ($p < 0.05$) pH value in the uppermost soil mineral (0–20 cm depth) layer. Furthermore, compared with the studied Arenosols, the concentrations of TN, Ca and Mg (only in site Lv2) were 53%–73%, 96%–793% and 460%–545% higher, respectively, in the studied Luvisols.

**Table 2.** Chemical properties of the mineral or peat top layers (0-20 and 20-40 cm depth) in the selected silver birch (*Betula pendula* Roth) forest stands.

| Soil Layer | Histosol (site Hs1) | Arenosol (site Ar1) | Arenosol (site Ar2) | Luvisol (site Lv1) | Luvisol (site Lv2) |
|---|---|---|---|---|---|
| | | | $pH_{CaCl2}$ | | |
| 0–20 cm | 4.25 ± 0.64 b | 3.63 ± 1.89 a | 3.26 ± 0.05 a | 3.29 ± 0.06 a | 5.37 ± 0.18 c |
| 20–40 cm | 5.29 ± 1.17 b | 4.30 ± 1.96 ab | 4.64 ± 0.58 a | 3.89 ± 0.61 a | 6.53 ± 0.30 b |
| | | | SOC, % | | |
| 0–20 cm | 30.80 ± 0.51 e | 1.62 ± 0.15 a | 2.99 ± 0.19 c | 3.81 ± 0.49 d | 2.66 ± 0.15 b |
| 20–40 cm | 28.67 ± 1.20 b | 0.48 ± 0.14 a | 0.49 ± 0.02 a | 0.59 ± 0.19 a | 0.42 ± 0.03 a |
| | | | TN, % | | |
| 0–20 cm | 2.42 ± 0.14 c | 0.17 ± 0.03 a | 0.15 ± 0.03 a | 0.26 ± 0.03 b | 0.26 ± 0.01 b |
| 20–40 cm | 2.24 ± 0.05 d | 0.08 ± 0.02 c | 0.04 ± 0.00 a | 0.08 ± 0.01 c | 0.05 ± 0.00 b |
| | | | $P_2O_5$, g kg$^{-1}$ | | |
| 0–20 cm | 0.044 ± 0.004 c | 0.058 ± 0.010 c | 0.171 ± 0.024 d | 0.016 ± 0.003 a | 0.026 ± 0.002 b |
| 20–40 cm | 0.024 ± 0.002 b | 0.130 ± 0.019 d | 0.067 ± 0.012 c | 0.011 ± 0.003 a | 0.126 ± 0.025 d |
| | | | $K_2O$, g kg$^{-1}$ | | |
| 0–20 cm | 0.118 ± 0.003 d | 0.028 ± 0.004 a | 0.042 ± 0.005 b | 0.035 ± 0.005 a | 0.057 ± 0.005 c |
| 20–40 cm | 0.028 ± 0.005 c | 0.014 ± 0.002 b | 0.013 ± 0.012 b | 0.009 ± 0.000 a | 0.057 ± 0.030 d |
| | | | Ca, g kg$^{-1}$ | | |
| 0–20 cm | 10.913 ± 0.284 e | 0.266 ± 0.025 b | 0.211 ± 0.028 a | 0.523 ± 0.133 c | 2.110 ± 0.165 d |
| 20–40 cm | 11.044 ± 1.047 d | 0.171 ± 0.018 a | 0.199 ± 0.035 a | 0.514 ± 0.005 b | 1.788 ± 0.224 c |
| | | | Mg, g kg$^{-1}$ | | |
| 0–20 cm | 1.117 ± 0.011 c | 0.083 ± 0.009 a | 0.070 ± 0.011 a | 0.076 ± 0.005 a | 0.382 ± 0.049 b |
| 20–40 cm | 1.081 ± 0.019 c | 0.061 ± 0.006 a | 0.063 ± 0.014 a | 0.062 ± 0.007 a | 0.438 ± 0.064 b |

**Note:** Results are expressed as the mean ± SE. The different lowercase letters indicate significant differences at $p < 0.05$.

### 2.3. Collection of the Silver Birch (Betula pendula Roth) Sap and Chemical Analyses

The samples of silver birch sap were collected from 10 March to 1 April 2017. A tapping device (patent no. LT 5813 B) was used for the collection of birch sap (Figure 2).

The holes (diameter 22 mm) were drilled in selected 20 trees growing in forest interior from 5 study sites at the height of 30 cm above the soil surface using a cordless drill and a feather drill bit. The tapping device included an implant (1) that was driven into a borehole using a rubber hammer. The implant was connected to a plastic tube (3) using a custom-made screw-on connector (2). The plastic tube was connected to a 10 litre plastic container (6) using a custom-made screw-on container lid, which had a 1 mm hole for airflow (4). This setup prevented insects and debris from contaminating the birch sap.

The chemical composition of the fresh birch sap was determined immediately after the collection of the samples. The physical and chemical parameters of the birch sap were analysed at the Laboratory of Biochemistry and Technology, Institute of Horticulture, Lithuanian Research Centre for Agriculture and Forestry (LAMMC). The ascorbic acid content in the sap was measured using titration with 2,6-dichlorophenolindophenol sodium salt solution [26]. The total soluble solids content was detected refractometrically using an Atago PR32 digital refractometer (Atago Co. Ltd., Tokyo, Japan) and expressed as ° Brix. The monosaccharide and saccharose contents in the sap were determined by the Bertrand method [27]. The electrical conductivity was measured with an ECTestr 11+ conductivity meter (Oakton, Vernon Hills, USA). The pH of the birch sap was measured by a CyberScan pH 6500 pH meter (Eutech Instruments).

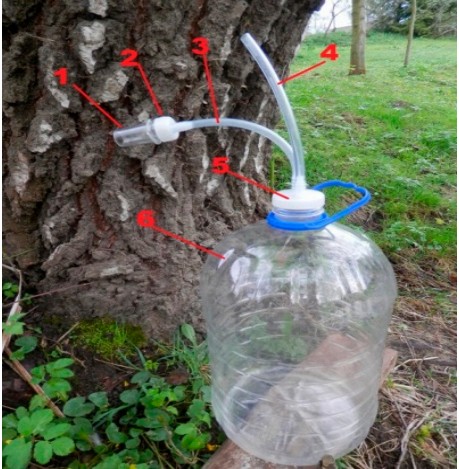

**Figure 2.** Tapping device components: (**1**) implant, (**2**) screw-on connector, (**3**) plastic tube, (**4**) plastic tube for air flow, (**5**) screw cap, and (**6**) container.

### 2.4. Statistical Analyses

The differences between quantity and biochemical composition of birch sap in different soils were compared using one-way analysis of variance (ANOVA) and Tukey's tests, with a significance level of 0.05. The influence of the dendrometric parameters on the birch sap quantity was analysed using linear regression. The model performance was evaluated using the determination coefficient ($R^2$). The statistical significance of the linear regression model was assessed using the F-test at a significance level of $\alpha = 0.05$. Principal component analysis was used evaluate the influence of the dendrometric parameters (H and DBH), soil group (Hs1, Ar1, Ar2, Lv1 and Lv2) and soil nutrient content (the mean concentrations of Ca and Mg determined in soil organic and mineral 0–40 cm depth layer) on the birch sap quantity and chemical composition. PCA analysis was based on the correlation matrix between the components and standardized variables.

## 3. Results

### 3.1. The Quantity of Sap Collected from Silver Birch (Betula pendula Roth) Trees with Different Dendrometric Parameters

The mean quantity of sap collected during the entirety of the exudation period (10 March–1 April) of silver birch (*Betula pendula*, hereafter, birch) from trees of different dendrometric parameters are presented in Figure 3.

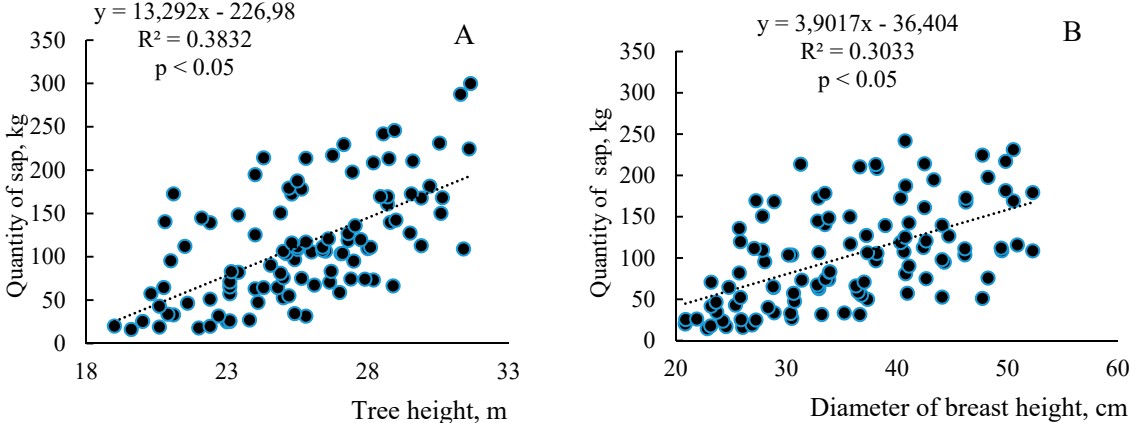

**Figure 3.** The quantity (kg) of sap collected from silver birch (*Betula pendula*) trees ($n = 103$) of different heights (**A**), diameters at breast height (**B**).

The mean quantity of birch sap increased with increasing dendrometric parameters, such as tree height (H) and diameter at breast height (DBH) (Figure 3). The highest mean quantity of birch sap was collected from trees 32 m in height, with a diameter at breast height of 50 cm.

*3.2. The Quantity and Biochemical Composition of Silver Birch (Betula pendula Roth) Sap Collected from Trees Growing in Different Soils*

The influence of different physical and chemical properties of the studied soils on the birch sap quantity and biochemical composition is presented in Figures 4 and 5 and Table 2.

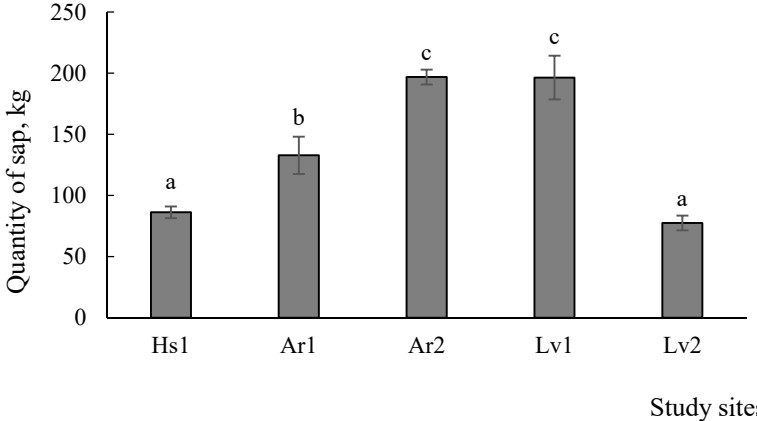

**Figure 4.** The mean quantity of sap harvested from silver birch (*Betula pendula*) forest stands growing in different soils. Abbreviations: Hs1—Histosol; Ar1 and Ar2—Arenosol; Lv1 and Lv2—Luvisol. The results are expressed as the mean ± SE, and different lowercase letters indicate significantly different ($p < 0.05$) means based on Tukey's test.

The results presented in Figure 4 indicate that the highest ($p > 0.05$) birch sap quantity (200 kg on average) was extracted from birch forest stands growing in Luvisol (site Lv1) and Arenosol (site Ar2) of normal humidity. An intermediate quantity (34% lower) of birch sap was collected from birch stands growing in temporarily flooded Arenosol (site Ar1). The lowest ($p < 0.05$) quantity (60% lower) of birch sap was collected in forest stands growing in undrained peatland (Histosol; site Hs1) and temporarily flooded Luvisol (site Lv2).

Soils with higher moisture content remain frozen longer in early spring. This could have had a significant effect on the sap quantity collected from forest stands growing in soils with higher moisture content (i.e., the Arenosol at site Ar1, Luvisol at site Lv2 and Histosol at site Hs1).

The obtained results presented in Figure 6 show the differences in the physical and biochemical properties of the birch sap collected from the forest stands growing in Histosols, Luvisols and Arenosols.

As shown in Figure 5, the average pH value varied from 6.1 to 6.4 units in all the studied birch sap samples. The birch sap mostly consisted of monosaccharides (0.7–1.0 g 100 g$^{-1}$), saccharose (0.2–0.4 g 100 g$^{-1}$) and ascorbic acid (5.7–6.3 mg 100 g$^{-1}$). The total saccharide content, expressed as the sum of the monosaccharide and saccharose concentrations, varied from 0.81% to 1.4% in studied birch sap samples. The results shown in Figure 5 indicate that the highest ($p < 0.05$; 50%–75% higher on average) total saccharide content was found in the birch sap samples collected from forest stands growing in temporarily flooded Luvisol (site Lv2). The intermediate value (0.9%) of the total saccharide content of the birch sap was found in undrained peatland (in Histosol at site Hs1); however, the lowest value (0.84% on average) was found in the birch sap collected from stands growing in Arenosols (sites Ar1 and Ar2) and a Luvisol with normal moisture conditions (site Lv1).

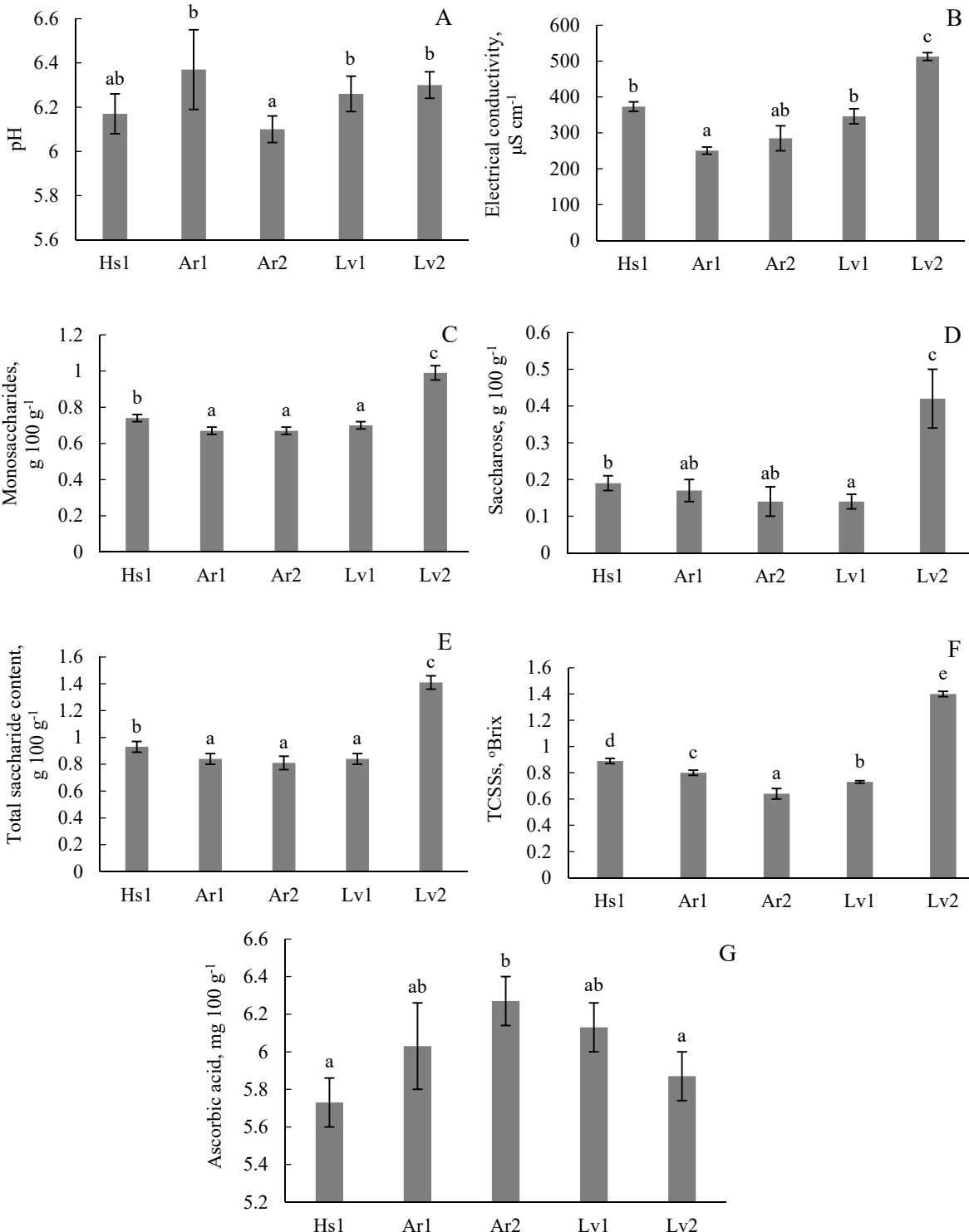

**Figure 5.** Mean pH value (**A**), electrical conductivity (**B**) and mean concentrations of monosaccharides (**C**), saccharose (**D**), total saccharide content (**E**), total content of soluble solids (TCSSs) (**F**) and ascorbic acid (**G**) in the birch sap of silver birch (*Betula pendula*) trees (in total, *n* = 103) growing in different soils. The results are expressed as the mean ± SE, and the different lowercase letters indicate significantly different (*p* < 0.05) means based on Tukey's test.

Consequently, the highest (*p* < 0.05) concentrations of monosaccharides and saccharose were found in the birch sap harvested from forest stands growing in temporarily flooded Luvisol (site Lv2), while the intermediate concentrations of monosaccharide and saccharose were found in the birch

forest stands growing on undrained peatland (Histosol at site Hs1). However, the lowest ($p < 0.05$) concentrations of monosaccharide and saccharose were found in birch forest stands growing on Arenosols (sites Ar1 and Ar2) and Luvisol with a normal moisture content (site Lv1).

The highest ($p < 0.05$) concentration of ascorbic acid (6.3 mg 100 $g^{-1}$ on average) was found in the sap collected from birch trees growing in Arenosol (site Ar2) with a normal moisture content. However, the lowest concentration of ascorbic acid was found in the sap from birch trees growing in temporarily flooded Luvisol (site Lv2) and undrained peatland (Histosol at site Hs1).

In the studied birch sap samples, the electrical conductivity may depend on the total content of soluble solids (TCSSs). The highest ($p < 0.05$) values of electrical conductivity (512 $\mu$S $cm^{-1}$) and TCSSs (1.4° Brix) were found in the sap samples collected from birch forest stands growing in temporarily flooded Luvisol (site Lv2). In comparison with the Luvisol (site Lv2), electrical conductivity was 72% lower ($p < 0.05$) and the TCSSs was 65% lower ($p < 0.05$) in the birch sap samples collected from forest stands growing in undrained peatland (in Histosol at site Hs1).

We found no relation between the pH of the sap and pH of soil samples (Table 3). However, there was a strong, positive relationship between the soil pH value and sap biochemical composition (monosaccharide, sucrose and total saccharide content) and electrical conductivity ($p < 0.05$). Furthermore, the soil pH ($r = 0.7$, $p < 0.05$) and soil Ca and Mg ($r = 0.5$, $p < 0.05$) were positively correlated with sap electrical conductivity and negatively correlated with ascorbic acid ($r = -0.5$, $p < 0.05$). In addition, the concentration of macronutrients (Ca and Mg) had a positive influence on the electrical conductivity ($r = 0.5$, $p < 0.05$) and biochemical composition (except for the concentrations of ascorbic acid) of the birch sap.

**Table 3.** Correlation matrix (based on Pearson correlation coefficients) between the pH values as well as nutrient concentrations (in the 0–40 cm soil layer) of studied soils and the pH values, electrical conductivity and biochemical compounds of silver birch (*Betula pendula*) sap samples.

| | Monosacharides | Saccharose | Sweetness | pHsap | Ascorbic Acid | Conductivity |
|---|---|---|---|---|---|---|
| pHsoil | 0.7* | 0.6* | 0.7* | 0.1 | −0.5* | 0.7* |
| OC | 0.0 | 0.0 | 0.0 | −0.1 | −0.2 | 0.0 |
| N | 0.0 | 0.0 | 0.0 | 0.2 | −0.3 | 0.0 |
| $P_2O_5$ | 0.1 | 0.0 | 0.1 | 0.0 | 0.0 | 0.0 |
| $K_2O$ | 0.3 | 0.2 | 0.3 | 0.0 | 0.2 | 0.3 |
| Ca | 0.5* | 0.4* | 0.5* | 0.0 | −0.3 | 0.5* |
| Mg | 0.5* | 0.5* | 0.5* | 0.0 | −0.3 | 0.5* |

Notes: * Significant Pearson correlation ($r$, $p < 0.05$). Abbreviations: pHsoil—pH in the soil, SOC—soil organic carbon, TN—total nitrogen, P₂O5—mobile phosphorus, K₂O—mobile potassium, Ca—mobile calcium, Mg—mobile magnesium, and pHsap—pH in the birch sap.

*3.3. The Influence of Dendrometric Parameters, Soil Group and Nutrient Status on Silver Birch (Betula pendula Roth) Sap Quantity and Biochemical Composition*

The data obtained in our study led us to evaluate the effect of the different soil nutrient properties and dendrometric parameters of the studied trees on the physical and biochemical properties of the birch sap (Figure 6).

The most correlated variables as dendrometric parameters of studied birch stands (H and DBH), soil type (Histosol (Hs1), Luvisols (Lv1 and Lv2), Arenosols (Ar1 and Ar2)), pH value and macronutrients (Ca, Mg) concentrations in studied soils, as well as pH value, electrical conductivity and biochemical composition (TCSSs, TSC, monosaccharides and saccharose) of studied birch sap were included in principal components analysis. The results presented in Figure 6 show that the quantity of birch sap is strongly correlated with the mentioned dendrometric parameters of studied birch trees. Meanwhile, the biochemical composition of the birch sap correlated with studied soil types, soil pH value and macronutrients as Ca and Mg.

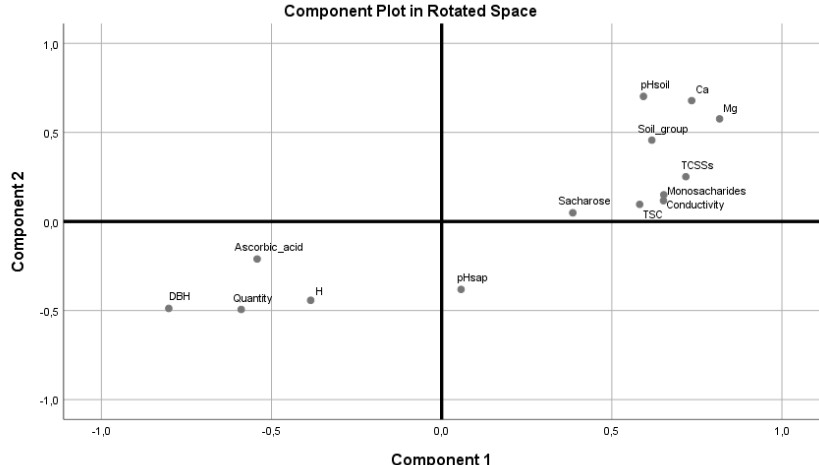

**Figure 6.** Principal component analysis for the silver birch (*Betula pendula*) sap physical and biochemical properties, the dendrometric parameters of the studied birch trees and the macronutrients (Ca and Mg) found in the different soils. Abbreviations: Quantity—quantity of the birch sap, pHsap—pH value of the birch sap, TSC—total saccharide content, TCSSs—total content of soluble solids, Conductivity—electrical conductivity; H—tree height, DBH—diameter at breast height, pHsoil—pH value of the soil, and Soil-group—type of studied soils (Histosol, Luvisol and Arenosol). Component 1 explains 55% of variance in data set; Component 2 explains 13%.

As shown in Figure 6, studied soil types had a strong influence on soil pH value, and the concentrations of Ca and Mg determined in the rhizosphere horizon (0–40 cm depth soil layer). Furthermore, soil type highly correlated with TCSSs, moderately correlated with the concentrations of monosaccharides and weakly with electrical conductivity of the birch sap. The concentrations of saccharose and TSC and TCSSs, as well as electrical conductivity of the birch sap, were moderately correlated with the nutrients (Ca and Mg) in the rhizosphere horizon, while the relation between the concentration of saccharose and mentioned soil nutrients was weak. The results of this study indicate that dendrometric properties such as tree height and diameter at breast height was a major factor that had an influence on the quantity of the birch sap. However, the sweetest and most nutritious (according to the value of the TCSSs) sap was collected from birch trees growing in nutrient-rich Histosol (undrained peatland, site Hs1) and temporarily flooded Luvisol (site Lv2) (see Table 2 and Figure 5).

## 4. Discussion

Birch sap ascends from roots to leaves through the xylem due to physical forces, pressure and osmotic gradients [15,19,30–32]. Positive root pressure causes early spring sap flow in birch trees [4]. The roots of trees with thick and tall trunks usually take up a larger area, allowing them to obtain more water from the soil.

Studies investigating the sap volume fluxes of birch species have been performed in 50- to 70-year-old forest stands [2,10,33]. However, the relationship between the age of the forest stands and the extracted sap volume has not been emphasized. Diameter at breast height (DBH) and sapwood depth are typically used as scaling parameters because they are usually positively related to sap velocity [16,34–36].

A recent study performed by Zajączkowska et al. [15] showed that the quantity of birch sap depended on different light availability in the forest stands. In our case, the dendrometric parameters of studied birch trees were similar. For example, the height (H) of the trees in each studied forest stand varied by approximately 2%–5% and diameter at breast height (DBH), by 2%–8% (Table 1). Hence, birch trees may have similar light requirements for photosynthesis in all the studied forest stands.

In general, the biochemical composition of birch sap is well documented. In many cases, the biochemical composition of birch sap found in our study was similar to results published in other studies. For example, Zajączkowska et al. [15] found that sugar content varied from 0.25% to 2.25% in silver birch sap. Kūka et al. [3] reported that silver birch sap consisted mostly of fructose (5.39 g 100 g$^{-1}$), glucose (4.46 g l00 g$^{-1}$) and sucrose (0.58 g 100 g$^{-1}$), while the average mean concentration of ascorbic acid was only 3.2 mg L$^{-1}$, and the average mean concentrations of $Ca^{2+}$ and $K^+$ were 53 and 41 mg L$^{-1}$, respectively. Kallio et al. [28] reported that the pH of birch sap varied between 5.5 and 8 units and that the average TCSSs was 0.5°–1.8° Brix. Furthermore, TCSSs consist of sugars (glucose and fructose), macronutrients (potassium (K), magnesium (Mg) and calcium (Ca)), acids (malic acid, succinic acid and others), free amino acids (citrulline, glutamine, and asparagine) and a wide variety of enzymes. Sonneveld and Voogt [29] reported that cations ($Ca^{2+}$, $Mg^{2+}$ and K+) had the most significant effect on electrical conductivity measurements. Therefore, the concentrations of macronutrients such as $Ca^{2+}$, $Mg^{2+}$ and $K^+$ may have the highest influence on the electrical conductivity of the birch sap collected from trees growing in different soils in our study.

The influence of soil macro- and micro-nutrient composition on the nutrients and bioactive compounds in silver birch (*Betula pendula*) and downy birch (*Betula pubescens*) sap was studied by Ozolinčius et al. [18] and Grabek-Lejko et al. [8]. Grabek-Lejko et al. [8] reported that the concentrations of macronutrients, such as Ca, Mg and K, in soil increased the concentrations of these nutrients in birch sap. However, contrary to our study, Grabek-Lejko et al. [8] did not find a relationship between the concentrations of macronutrients (Ca, Mg and K) in the soil and the concentrations of glucose, fructose and sucrose in the sap. Ozolinčius et al. [18] stated that higher concentrations of monosaccharides and sucrose were found in sap harvested from silver birch trees growing in temporarily flooded Cambisols and that, in contrast, lower concentrations of mentioned biochemical compounds were found in the sap of trees growing in Arenosols with normal moisture content. This study confirmed our findings that the sweetest sap was extracted from birch trees growing in nutrient-rich soils.

Numerous studies have investigated the influence of soil nutrient status on the biochemical composition of tree sap in sugar maple forest stands. Wild and Yanai [37] investigated maple sap sugar content after soil fertilization with N, P, or Ca and found that maple trees with higher sugar concentrations in their sap were growing in sites with higher soil nitrogen mineralization. Moreover, the combination of soil macronutrients, such as K, Ca, and Mg, increased the sweetness of sap from trees in northern Vermont in North America [38]. Costanza-Robinson et al. [39] found positive but statistically insignificant correlations between cations (Ca, Mg, and Mn) found in the soil and maple sap. This study did not find a relation between Ca in the soil and sap sugar content. However, the results were affected by the relatively low number of studied sites.

Laing and Howard [13] reported that healthier trees with larger crowns and greater growth rates tend to have sweeter sap. Furthermore, sugar maples growing in soil with lower Ca and Mg contents are often not as healthy as those growing in soils with higher cation contents [40–42]. Liming, the addition of Ca and Mg to soils, decreased the symptoms of maple decline [43–45]. This phenomenon can also be explained by indirect effects, as Ca and Mg lower soil acidity and simultaneously reduce the toxicity of Al and Mn cations to tree roots [46]. Safford [47] found that the DBH of sugar maple increased by 19% after lime addition and by 2-fold after lime plus NPK addition.

## 5. Conclusions

The quantity and physical and biochemical properties of birch sap were analysed from over-mature 73- to 105-year-old silver birch (*Betula pendula* Roth) stands growing in Histosol, Luvisol and Arenosol soils with different moisture and nutrient content. The studied silver birch trees had different dendrometric parameters: the height of trees varied between 19–32 m and diameter at breast height varied from 20 to 50 cm. The results of this study showed that the most productive silver birch trees for sap harvesting were taller than 28 m and had a diameter at breast height over 40 cm. Furthermore, compared with flooded soils (Histosol and Luvisol (Lv2)), higher quantity birch sap was collected

from birch trees growing in well-aerated soils (in Luvisol and Arenosol with normal moisture content). However, the concentrations of Ca and Mg in the rhizosphere horizon (soil organic or mineral layer of 0–40 cm depth) may have had a positive effect on the saccharose, monosaccharide and total saccharide levels, as well as the total content of soluble solids in the studied birch sap. The results highlight that the sweetest sap is collected from birch stands growing in nutrient-rich organic (undrained peatland Histosol) and mineral soils (Luvisols). Further research is now needed to determine the number of consecutive years that birch trees can be tapped without compromising tree health and to investigate the diseases caused by pathogens and the volume of nonconductive wood associated with taphole wounds.

**Author Contributions:** J.M., V.V., E.B., and K.A. conceived and designed the study; J.M. and P.V. performed the experiments; D.Č. analysed the data and wrote the paper; and K.A. reviewed and edited the paper. All authors have read and agreed to the published version of the manuscript.

**Funding:** This study was financed by the Vytautas Magnus University Agriculture Academy and by the Institute of Horticulture, Lithuanian Research Centre for Agriculture and Forestry. The work is partly attributed to the technological development project through a contract with the Agency for Science, Innovation and Technology Nr.31V-58.

**Conflicts of Interest:** The authors declare no conflict of interest.

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
