# Peer review of "The Quantity and Biochemical Composition of Sap Collected from Silver Birch (Betula pendula Roth) Trees Growing in Different Soils"

_forests, doi:10.3390/f11040365_

Round 1
Reviewer 1 Report
This is an interesting study, and one that should be of interest to the readers of Forests. The introduction is quite enlightening, and leads to a clear statement of purpose and 3 objectives -- which are a little redundant. I think objective 3 is to investigate the interaction of dendrometric parameters and soil chemical properties -- slight restatement
The Methods section would benefit from a bit more information -- some simple, some not so simple. For example, are these pure stands of silver birch? Are there other tree species present? Are they all of the same stand density (number of trees) and composition? The first time an abbreviation is used, it should be spelled out -- it took me awhile to find a definition of CBH for instance.
The description of statistical analysis was very brief, and I'm not sure that the results actually derive from the statistical analyses. For example. The authors state that the mean quantity of birch sap depended somewhat on tree age citing Figure 3 -- but Figure 3 only compares the quantity of sap among the 5 sites, which happen to have different ages. These should be the same values as in Figure 5, I think?
There is no indication whether the linear regression models in Figure 4 are statistically significant or not. The R2 is just a value of goodness of fit -- and these are relatively low values.
Section 3.2 There is a significant amount of reasoning that either should be in the discussion section, or possibly in the Introduction or methods section. This holds for the paragraph about light availability (first para), and the sections starting in lines 236 .
Line 245. Why do you assume that "concentrations of the above mentioned macronutrients have the highest influence on electrical conductivity of the birch sap collected..." -- This is a conclusion, not a result and goes in the Discussion section.
Finally, the Principal Component analysis is not really explained or, quite frankly explored. As it is now, the results are very general, and not obviously related to this Figure 7 or to the statistical analysis of it.
In the discussion there are numerous references to "this study", and it is not clear which study is being referred to. This is a minor point, but needs clarification so that your readers are not confused.
In the conclusions, there are statements made that have not been demonstrated using your result, "...concentrations of Ca and Mg in the rhizosphere horizon may have had a positive effect on ...." I don't think you demonstrated this with the results of your statistics.
Author Response
Authors’ responses to referees’ comments on:
Manuscript ID: forests-736820_ The quantity and biochemical composition of sap collected from silver birch (Betula pendula Roth) trees growing in different soils
Authors: Justas Mingaila, Dovilė Čiuldienė, Pranas Viškelis, Edmundas Bartkevičius, Vladas Vilimas, Kęstutis Armolaitis
We are grateful to both referees for their generally supportive comments on this work, and for agreeing to review a manuscript. We have considered comments and technical corrections when producing the revised version of manuscript. Responses to these comments are now provided.
Referee 1
Comment/ response No. |
Comments |
Responses |
1 |
This is an interesting study, and one that should be of interest to the readers of Forests. The introduction is quite enlightening, and leads to a clear statement of purpose and 3 objectives -- which are a little redundant. I think objective 3 is to investigate the interaction of dendrometric parameters and soil chemical properties -- slight restatement |
Objectives 1 and 2 deleted and remained objective “to evaluate how dendrometric parameters of studied pure silver birch trees and soil chemical properties influence sap quantity and biochemical composition”.
|
2 |
The Methods section would benefit from a bit more information -- some simple, some not so simple. For example, are these pure stands of silver birch? Are there other tree species present? Are they all of the same stand density (number of trees) and composition? The first time an abbreviation is used, it should be spelled out -- it took me awhile to find a definition of CBH for instance. |
Our study was performed in pure silver birch stands. The dendrometric parameters as tree density per ha and stand volume was added In Table 1. We removed CBH from results, because CBH is the part of tree height.
|
3 |
The description of statistical analysis was very brief, and I'm not sure that the results actually derive from the statistical analyses. For example. The authors state that the mean quantity of birch sap depended somewhat on tree age citing Figure 3 -- but Figure 3 only compares the quantity of sap among the 5 sites, which happen to have different ages. These should be the same values as in Figure 5, I think? |
In 2.4 section the more information about used statistical analyses and methods was added. We agree with this comment, therefore we removed Figure about sap quantity extracted in different age birch stands. |
4. |
There is no indication whether the linear regression models in Figure 4 are statistically significant or not. The R2 is just a value of goodness of fit -- and these are relatively low values.
|
p value was added in figure 3 A and B sections. |
5. |
Section 3.2 There is a significant amount of reasoning that either should be in the discussion section, or possibly in the Introduction or methods section. This holds for the paragraph about light availability (first para), and the sections starting in lines 236 .
|
Information was removed from 3.2 section and added to discussion section, line 292 |
6. |
Line 245. Why do you assume that "concentrations of the above mentioned macronutrients have the highest influence on electrical conductivity of the birch sap collected..." -- This is a conclusion, not a result and goes in the Discussion section.
|
This information was removed from 3.2 section and added to discussion section, line 309.
|
7. |
Finally, the Principal Component analysis is not really explained or, quite frankly explored. As it is now, the results are very general, and not obviously related to this Figure 7 or to the statistical analysis of it. |
The more detail information of principal component analysis was added in 2.4 section. Also we did major changes in describing results (3.3 section). |
8. |
In the discussion there are numerous references to "this study", and it is not clear which study is being referred to. This is a minor point, but needs clarification so that your readers are not confused. |
We changed with words “in our study” or “in our case”. We tried to avoid misunderstanding with words “this study”. |
9. |
In the conclusions, there are statements made that have not been demonstrated using your result, "...concentrations of Ca and Mg in the rhizosphere horizon may have had a positive effect on ...." I don't think you demonstrated this with the results of your statistics |
We think our results led us to conclude so, because we studied soil chemical properties in 0-40 cm layer which is the major part of rhizosphere horizon (0-50 cm depth). |
CLOSING COMMENTS TO THE EDITOR:
Again, we appreciate the opportunity to revise our work for consideration for publication in Forests journal. We hope our revision meet your approval. We next detail our responses to referee’s concerns and comments. Thank you for taking the time and energy to help us improve the paper.

Reviewer 2 Report
Birch sap is recognized non wood forest product in many countries in the northern hemisphere. However, there are scarce data on the sap properties in relation to tree age, volume or forest stands characteristics including soil. In submitted paper, the above-mentioned properties were analyzed in five different silver birch stands and two localities in Lithuania. The current study evidenced some trends and relationships between trees characteristics, soil properties and sap quantity and biochemical composition. That's why in my opinion this paper is interesting for publication.
Unfntunterlly I find also some weaks points of the study listed briefly in following points:
- The main objectives of the study were in points 1 and 2 and then summarized in objective 3. This is partly overlapped with point 3 and in my opinion the sentence “to evaluate how both dendrometric parameters and soil chemical properties influence sap quantity and biochemical composition” is complex and enough precise influence the aims of the study. I suggest to delete objective 1 and 2.
- Material is not clear described but is essential to understand the results. I can not find information about 103 studied trees vere distribution in the studied stands. Please describe it because it is essential for the results.
- It will be interesting to provide additional characteristic about studied stands like tree density on hectars, type of terrain cover by trees, bonitations and finally descrive more precisely (using forest terminology) moisture levels.
- Crown base height is rather parameters describing the crown height and the tree potential vitality. I suggest replace it by crown height (tree height minus crown base height).
- Statistical analysis are described very briefly. I would like to see more detail information about model of Anova and principal component analysis. For example were the values scaled or not, standardised, what type of method was used etc..?
- Fig 4. I suggest to combine part A and B in the one figure using volume instead of height and DBH. Height and DBH are highly correlated. Additionally there is lack of the p value on particular graph and description of X axis “height”..
- Please do not mix result with discussion. Lines 184-192 have to be moved to discusion.
- I do not understand why authors cite (personal communications) for description of the studied stands? In fact it was visited and sampled trees are located inside the study area. So, this knowledge have to be available from direct observations.
Above mentioned disadvantages are easy to correct and after this the paper can be published. The results indicated for relationship between birch sap characteristics and soil is interesting findings.
Author Response
Authors’ responses to referees’ comments on:
Manuscript ID: forests-736820_ The quantity and biochemical composition of sap collected from silver birch (Betula pendula Roth) trees growing in different soils
Authors: Justas Mingaila, Dovilė Čiuldienė, Pranas Viškelis, Edmundas Bartkevičius, Vladas Vilimas, Kęstutis Armolaitis
We are grateful to both referees for their generally supportive comments on this work, and for agreeing to review a manuscript. We have considered comments and technical corrections when producing the revised version of manuscript. Responses to these comments are now provided.
Referee 2
Comment/ response No. |
Comments |
Responses |
1 |
The main objectives of the study were in points 1 and 2 and then summarized in objective 3. This is partly overlapped with point 3 and in my opinion the sentence “to evaluate how both dendrometric parameters and soil chemical properties influence sap quantity and biochemical composition” is complex and enough precise influence the aims of the study. I suggest to delete objective 1 and 2 |
Objectives 1 and 2 deleted and remained objective “to evaluate how dendrometric parameters of studied pure silver birch trees and soil chemical properties influence sap quantity and biochemical composition”.
|
2 |
Material is not clear described but is essential to understand the results. I can not find information about 103 studied trees vere distribution in the studied stands. Please describe it because it is essential for the results |
In 3.2 section, 135 line we added information that a we selected 20 trees growing in forest interior in 5 study sites. |
3 |
It will be interesting to provide additional characteristic about studied stands like tree density on hectars, type of terrain cover by trees, bonitations and finally descrive more precisely (using forest terminology) moisture levels |
In section 2.1, line 83 we added information that selected birch stands were growing in plains. In Table 1. the information about tree density per ha and stand volume was added. Unfortunately we did not measure soil moisture, but soil moisture condition was identified in all study sites. |
4. |
Crown base height is rather parameters describing the crown height and the tree potential vitality. I suggest replace it by crown height (tree height minus crown base height).
|
According to Referee`s comment that CBH and H are correlated variable, we removed CBH from results and PCA model |
5. |
Statistical analysis are described very briefly. I would like to see more detail information about model of Anova and principal component analysis. For example were the values scaled or not, standardised, what type of method was used etc..?
|
In 2.4 section the more information about used statistical analyses and methods was added. |
6. |
Fig 4. I suggest to combine part A and B in the one figure using volume instead of height and DBH. Height and DBH are highly correlated. Additionally there is lack of the p value on particular graph and description of X axis “height”..
|
CBH value was removed. Also p value was added in figure 3 A and B sections. |
7. |
Please do not mix result with discussion. Lines 184-192 have to be moved to discusion |
This information was moved to discussion |
8. |
I do not understand why authors cite (personal communications) for description of the studied stands? In fact it was visited and sampled trees are located inside the study area. So, this knowledge have to be available from direct observations.
|
We added needed information about tree density in Table 1. Also we removed all (personal communication) from manuscript. |
Round 2
Reviewer 1 Report
I think the authors have addressed the major concerns raised in my earlier review, but now the ms needs significant text editing, particularly where text has been added.